# Heat Consumer Model for Robust and Fast Simulations of District Heating Networks Using Modelica

Johannes Zipplies *, Janybek Orozaliev, Ulrike Jordan  and Klaus Vajen

University of Kassel, Institute of Thermal Engineering, Department of Solar and Systems Engineering, Kurt-Wolters-Str. 3, 34125 Kassel, Germany
* Correspondence: solar@uni-kassel.de

**Abstract:** Dynamic thermo-hydraulic simulations of district heating networks (DHN) are essential to investigate novel concepts for their sustainable design and operation. To develop solutions for a particular case study, numerous long-term simulations are required. Therefore, computational effort for simulation is critical. Heat consumers (HC) are numerous and determine the dynamics of mass flows and return temperatures in the DHN. Thus, the way in which HCs are modeled has significant impact on the computational effort and the results of the simulation. This article presents a novel Modelica-based model for HCs that builds on an existing simplified modeling approach (open-loop design). The calculation of mass flow and return temperature is improved in terms of robustness, plausible behavior and low computational effort. In particular, the model reacts to limited differential pressure and supply temperatures to ensure plausible behavior across all operating conditions, including undersupply situations. The model is successfully tested using an exemplary DHN. The analysis proves that the HC model itself requires little time to simulate. Nevertheless, it significantly influences the simulation time for the entire DHN, which varies by a factor of five for the investigated system depending on the HC model. Fast dynamics, including a bypass in the model and correction of deviations between set point and actual heat load increase the simulation time, so users should sensibly choose how to use these options. HC models triggering many state events result in high computational effort. Compared to other simple HC models, the proposed model produces more plausible results while maintaining at least equal simulation performance (for models without bypass) or even improving it (for models with bypass, CPU time is reduced by at least 35%).

**Keywords:** modelica; district heating network; heat consumer; simulation performance



## 1. Introduction

In light of the man-made climate crisis, a fast decarbonization of heating has to be achieved. Within this transition of the heat sector, district heating is a recommended solution for densely populated areas, and it is expected to be expanded to supply around 50% of the heat demands in Europe in 2050 [1]. District heating facilitates a combination of various renewable heat sources, excess heat usage and heat storage (even seasonal) and coupling to the electricity sector to reach an economically viable sustainable heat supply system, the so-called fourth-generation district heating [2].

The transformation towards fourth-generation district heating requires the development and successful implementation of novel concepts for heat supply and distribution, which raises numerous challenges: The integration of distributed renewable heat sources leads to a more decentralized feed-in and requires reduced temperatures in the district heating network (DHN) [3]. This may entail new hydraulic bottlenecks [4], (more frequent) flow reversals, formation of cold plugs, increased thermo-mechanical stress on the pipes, frequent changes of the hydraulically critical path and temperature undersupply of heat consumers (HC) that are far from heat supply units or require high supply temperatures. The need for densification and expansion of the existing DHNs makes things even more difficult.

These challenges require targeted reinforcement of the DHN or even novel network layouts [4], as well as novel concepts for design and placement of new heat supply and heat storage units, and innovative operation strategies for all components. To develop these concepts, simulations of the DHNs are required [3,5] using dynamic thermo-hydraulic models able to handle bi-directional flows [6]. Furthermore, the models need to be suitable for long-term simulations, even annual, if heat sources show seasonal fluctuations or if long-term heat storage is to be considered. In addition, the models must be applicable for the simulation of large DHNs, as existing DHNs are to be examined. Thus, good simulation performance is essential.

### 1.1. Simulation of District Heating Networks: Tools and Applications

A variety of proprietary and open-source tools exists for the simulation of DHNs. Brown et al. [7] present an overview on commercial modeling tools and their capabilities, concluding that they are limited with respect to computational time, level of precision and scalability, so that more research is needed towards integrated district energy models. Soons et al. [8] provide a comprehensive comparison of potentially appropriate modeling environments for dynamic simulations of DHNs, namely Modelica (using Dymola), Matlab/Simulink and TRNSYS. They conclude that Modelica performs better concerning modularity, multi-domain modeling, realistic control behavior and flexibility, which are important features to facilitate the development and assessment of innovative systems designs and operation strategies. Similarly, Giraud et al. [9] conclude, in a comparison of simulation tools for DHNs, that Modelica's lower development effort and wider modeling possibilities overbalance the observed higher computational cost. Schweiger et al. [5] compare major multi-purpose tools and conclude that Modelica is best suited for the simulation of district energy systems with a limitation concerning the simulation of large scale systems. Other tools have no or limited suitability for power distribution systems and co-simulation (IDA ICE and TRNSYS) or limited suitability for district heating and building simulation (Simulink).

The simulation environment TRNSYS is used for the transient simulation of thermal components. Current studies examine substations for solar thermal feed-in into DHNs [10], complex heat supply systems with large solar thermal and seasonal storage [11] or small-sized DHNs and the connected buildings with the goal to find a minimal operation temperature at every instant [12].

Modelica is an open-source modeling language with two core features that make the language well suited for dynamic simulations of DHNs. First, Modelica is equation-based and acausal. This means that the models consist of equations—not assignments—so that there is no predefined causality [13]. In consequence, the interfaces between component models must not be categorized into in- and output (unlike in Matlab–Simulink and TRNSYS). In DHN simulations, the pipe model must be able to handle flow reversals in order for the connectors from one to the other pipe to alternately act as in- or output, which can be easily implemented in Modelica. This is necessary in order to ensure that compared to causal modeling approaches, significant effort by the user is avoided [5]. Second, Modelica and its model libraries are designed to build multi-physics models across all domains. For DHN simulations, thermodynamics as well as fluid dynamics and potentially the electrical domain (if sector coupling is to be examined) are needed and inherently supported by Modelica (unlike TRNSYS which focuses on thermal modeling) [8].

In accordance with its good suitability, Modelica is used in many studies on DHN. Annual simulations of a small DHN with 25 HCs and decentralized prosumers were carried out using Modelica in order to evaluate different scenarios [14]. Waste heat integration was evaluated for a small DHN [15], as well as a medium-sized DHN with about 100 HCs [16]. The latter article demonstrates that dynamic simulations allow for identification of the share of waste heat that can be integrated into an existing DHN and to analyze hydraulic bottlenecks and temperature undersupply that arises from lower supply temperatures. Neirotti et al. [17] evaluate the potential for lowering the temperatures in a small existing

DHN using Modelica. Using test cases with about 100 HCs, Schweiger et al. [18] and O'Donovan et al. [3] demonstrate that Modelica is well-suited for the simulation of medium-sized DHNs.

*1.2. Fast Simulation of District Heating Networks*

Simulation performance depends on a multitude of factors. In general, modeling of DHNs requires a reasonable compromise between accuracy (and thus the level of detail of the models) and simulation performance, with the optimal point depending on particular application [19]. Jorissen et al. [20] provide an analysis of simulation speed for building energy systems (including thermo-hydraulic systems). Important measures are to avoid algebraic loops (that may, for example, be introduced by the differential pressure control in a DHN model), inefficient code and to keep the number of evaluations as low as possible.

Aggregation is an important strategy to keep the simulation time of DHN models within reasonable limits. The so-called "German Method" [21] and "Danish Method" [22] are widely used aggregation algorithms. Larsen et al. [23] and Falay et al. [24] provide comparisons of the two methods, proving their ability to drastically reduce the computational effort while maintaining good accuracy of the results. However, both methods use a certain network state as a basis for the aggregation algorithm. Thus, the aggregation methods are valid for situations similar to that state, with the frequency of errors increasing the more the actual situation differs. Falay et al. [24] state that in the case of flow reversal, the aggregation is no longer valid and the procedure has to be repeated each time a flow reversal occurs somewhere in the DHN. Thus, the application of these aggregation algorithms to reduce computational effort is limited with respect to the current challenges in DHNs.

Boussaid et al. [25] apply a data-driven approach (graph neural network) as a surrogate model for DHN to be used in optimization tasks. This approach drastically reduces the simulation time compared to a physical model, but results with changed control parameters cannot be captured properly. Furthermore, the data-driven model does not compute and expose the physical states within the network, so that bottlenecks or temperature undersupply cannot be analyzed. Apart from that, data-driven models have an application range that is strictly limited to the range of operating conditions for which sufficient data exist, so the analysis of scenarios that extend beyond that is impossible.

*1.3. Heat Consumer Models*

In the literature, various HC models using Modelica are described. Giraud et al. [26] developed and validated a HC model with a heat exchanger model and a valve to control the primary mass flow. The model requires the secondary return temperature as an input, a quantity that is usually not known. Neirotti et al. [17] use a detailed HC model including component models for building heat loss and heat capacity, radiator, heat exchanger, pumps and control loops. Their simulation use case contains only nine HCs and the network model is reduced to only one pipe instance for flow and return line. Kauko et al. [14] present a HC model with heat exchangers, valves, regulators and even radiators to perform annual simulations of a small DHN without loops, reporting a simulation time of 3 h. Leitner et al. [27] present a Modelica HC model that includes separated heat exchangers for heating and domestic hot water, pumps, valves and radiators. Their annual simulation of a small simple radial DHN (14 HC) takes 4 to 10 h. A simplified HC model with instantaneous heat release and a balancing valve for mass flow is described by del Hoyo Arce et al. [19]. Stock et al. [16] use a strongly simplified HC model for Modelica that is intended to investigate waste heat integration and supply temperature reduction in large DHNs. However, their model neglects important effects (such as pressure undersupply) and seems to be too simple (constant bypass mass flow) (see Section 2.1.1). Brown et al. [7] mention that further research on how to model HC behavior in Modelica is required.

It is obvious from the literature that HCs in DHNs can be modeled at very different levels of detail. The exact implementation is an important factor for simulation performance for two reasons: First, the effort for simulating each of them is crucial for the overall

simulation time because within a DHN, the HCs are very numerous. Second, the HCs have a major impact on dynamics of mass flows and temperatures within the network and thus determine the effort to compute the fluid and temperature propagation in the pipes.

*1.4. Contribution of This Article*

Based on the literature review, we see a relevant research gap concerning the models of HC for long-term simulations of large DHNs. It includes two aspects:

- Known HC models are either too detailed (long simulation times, high effort for proper parameterization) or too simple (do not plausibly reflect the relevant effects).
- To the best of our knowledge, the importance and influence of HC model design on simulation time has never been evaluated in detail.

Given this research gap, this article aims to answer the following two research questions:

- How should a HC be modeled to yield plausible results at various operating conditions and enable robust and fast simulations of large DHNs?
- How and to what extent does the HC model design affect the overall simulation time?

The novel HC model presented in this paper enriches an existing approach for simplified HC modeling (open-loop, described in [16]) with features that ensure robust and plausible behavior of the model under various operating conditions (including undesirable situations, such as excessively low supply line temperatures or differential pressures) while keeping the computational effort for simulations as low as possible. The influence of the HC model on simulation time is evaluated using this novel HC model with different configurations and simple models from two other Modelica libraries.

## 2. Simulation of DHN Using Modelica

This section offers a brief overview on Modelica libraries used in this research article and explains important strategies for fast simulations of DHNs with respect to HC models.

*2.1. Models for DHN and HC*

The *Modelica Standard Library* [28] provides a large number of base models (such as the fluid connector) and component models for thermo-fluid systems. Moreover, van der Heijde et al. [29] developed and validated a dynamic plug-flow pipe model that is freely available via the *Modelica IBPSA Library* [30] and which is used within other libraries with models specialized in the simulation of DHN.

2.1.1. AixLib

*AixLib* is an open-source Modelica library for the simulation of energy systems on building to district scale developed at RWTH Aachen University [31–33]. It extends the *Modelica IBPSA Library* and has a section *DistrictHeatingCooling* with models specialized for the simulation of DHN.

Within this section, the library provides so-called "open-loop" models for HCs. This means that the models do not contain a fluid model that connects flow and return line, which allows for decoupling the respective equation systems for fluid flow and pressures in the supply and return line of the DHN. Stock et al. [16] state that open-loop models reduce the computational effort and yield valid results when the research focus is on heat distribution and not on control of the HCs or heat sources. They successfully evaluate the hydraulic effects of the integration of a waste heat source into an existing DHN at different temperature levels.

The HC models determine the required mass flow based on a heat load input and the temperatures. The return line temperature is either a constant value or set to achieve a constant temperature difference to the supply line. A bypass that maintains a minimum mass flow may be included. It is active whenever the HC mass flow drops below a threshold (irrespective of temperatures and pressures), sets the heat flow to zero and triggers state events whenever activated or deactivated.

The strength of *AixLib* is its broad scope that covers the whole area of heat supply systems for buildings. The district heating HC models, however, are very basic and might produce implausible results, e.g., if inappropriate load time series are used (including steps or excessively high peaks) or in undersupply situations (insufficient differential pressure is ignored, and if the supply temperature is too low, a constant temperature difference is used instead of the set point return temperature; see Section 4.2.5). Furthermore, the bypass models result in an unstable return temperature that switches at an undetermined moment (see Section 4.2.3). Nevertheless, the HC model proposed in this article builds upon the open-loop design of these models and tries to overcome the described weaknesses.

### 2.1.2. DisHeatLib

Leitner et al. [27] describe a method to assess the operation of coupled heat and power networks. The authors published their Modelica models within the *DisHeatLib* library [34], which builds upon the *Modelica IBPSA Library* and contains a variety of models for DHN Simulations.

To model HCs, the library provides models for *demand* (intended as a simple representation of a heat load) and *substation* (modeling heat transfer from the network to the HC). The *substation* models include a variety of technical configurations (with or without heat exchanger, optional heat storage and/or bypass), so that these technical options and their behavior within a DHN can be examined. However, all HC models in *DisHeatLib* include control loops, fluid models that connect supply to return line, and some have a high degree of detail as the various components are explicitly modeled, which results in high computational effort to simulate a DHN with numerous HCs. Thus, these HC models are useful to examine different substation configurations, but are not well suited for long-term simulations of large DHN, as intended in this contribution. Thus, the novel HC model uses a more simple approach, where the components are not modeled in detail.

### 2.1.3. DHNSim

At the Department of Solar and Thermal Engineering of the University of Kassel, Modelica models for long-term simulation of whole DHNs have been developed within the in-house *DHNSim* library. The pipe model in *DHNSim* builds upon the plug flow pipe model by van der Heijde et al. [29]. Furthermore, the library contains models for supply units, the HCs (described in this contribution; see Section 3) and the required environment to easily build a consistent DHN model. Zipplies et al. [35] present an overview on the structure, goals and general implementation of the models.

The strength of *DHNSim* are annual simulations of DHNs at low computational effort. The library is specialized for this application and currently has a limited number of component models. However, the models from *DHNSim* are compatible with those from many other libraries (including the *Modelica Standard Library*, *IBPSA Library*, *AixLib* and *DisHeatLib*), as the same connectors are used, so that suitable component models from these libraries may be combined with its models.

### 2.2. Strategies for Fast Simulations of DHN

The total CPU time of a simulation using variable step solvers can be approximated by a constant share for initialization and share that is the product of the number of steps and the time that is needed to compute one step [20]. Figure 1 illustrates a general consideration of the drivers for computational effort of DHN simulations. On the one hand, the pipe network model results in a large system of equations that has to be solved for each simulation step and numerous states to integrate. Thus, it determines the effort to calculate one simulation step. On the other hand, the models of the supply unit and HCs do not cause much computational effort themselves if they are simple. However, as they determine the mass flows, temperatures and pressures in the network (and their derivatives), they have a crucial impact on the number of steps that a variable step size solver has to calculate. Given

this consideration, the following subsections describe general strategies for fast simulations of DHN that apply to the proposed HC model, which is described in detail in Section 3.

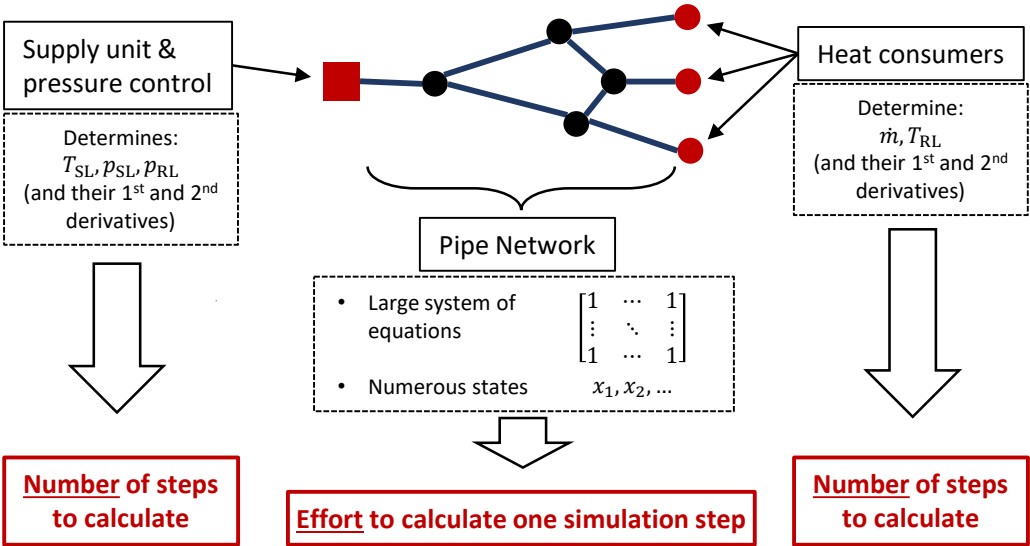

**Figure 1.** Drivers for computational effort of district heating network (DHN) simulations: while the model of the pipe network dominates the effort to calculate one step, models of the supply unit and even more of the heat consumers (HCs) determine the number of steps that the simulation requires.

### 2.2.1. Simplified Modeling Approach of HCs

The simulation of a large branched or even meshed pipe network is a complex task that requires high computational effort. Therefore, it is recommendable, or even absolutely necessary, to limit the degree of detail of the models of the supply unit and the HCs in the DHN to a minimum extent that still leads to valid results. This applies especially to the HCs, as they are numerous and determine the mass flows and return temperatures for the pipe network. Thus, the proposed HC model does not contain detailed physical models for the actual components of substation and secondary side (pipes, valves, heat exchangers, pumps, heat storage, radiators, floor heating, etc). This simplified modeling approach allows for following the open-loop design implemented in *AixLib* (see Section 2.1.1). The evaluation of the simulation performance in Section 4.2.7 clearly indicates that this is an important measure, as the HC model from *DisHeatLib* (which does not follow the open-loop design and contains a few more component models including a control loop) has a substantially higher simulation time compared to the equivalent open-loop models without bypass.

The HC model simply uses a prescribed heat flow (or, optionally, a mass flow) as input and uses the actual temperature in the supply line and a prescribed return line temperature (constant value or as additional variable input) to calculate the mass flow. While this seems to be a very simple modeling task at first glance, some more details and features are needed to obtain fast, stable and valid simulations with such HC model. These are described in Section 3 and include major improvements compared to the open-loop models in *AixLib*.

### 2.2.2. Avoiding Events

Simulation models may include equations or algorithms that abruptly change the model behavior. Examples are flow reversals (mass flow changes the sign) or switching units on and off (boolean variable changes the value). Within Modelica, these moments are called "events", and whenever the integration algorithm detects such an event, the integrator needs to determine the exact point of time when this abrupt change occurs and restart the simulation with the changed model behavior from this point so that the transition from one state to the other is simulated correctly.

While this approach avoids inaccurate results or even failures of the simulation that might occur otherwise, it also adds computational effort to the simulation [20]. Thus,

models should generate events only if necessary, and high numbers of events should be avoided in the use case of long-term simulations of large DHNs. Typical triggers for events in simulations of DHNs are flow reversals in pipes, switching units on and off (event may be avoided depending on unit model formulation, see example below) or reaching limits of controllers.

If a variable is continuous at an event, it is possible to prevent the event using the Modelica built-in function `smooth()` [13]. Furthermore, *Modelica Standard Library* provides the function `Modelica.Fluid.Utilities.regStep()` to approximate a step by smooth transition that is once continuously differentiable and prevents events [28]. Both functions are very useful to avoid events in the HC model and are used in its implementation wherever applicable.

To demonstrate the effectiveness of this approach, we consider the simple example for bypass operation in Figure 2a. The input temperature to a house lead-in pipe varies between 68 and 70 °C. The outlet temperature does not fall below 65 °C so that hot water is available at any time. This is achieved by a varying mass flow, that is, controlled in two ways: with an On/Off controller or with the bypass control block from *DHNSim* which uses the `regStep()` function to increase the mass flow within a bandwidth around the set-point temperature.

The simulation results in Figure 2b show that both implementations succeed in keeping the temperature at the pipe outlet close to 65 °C. In contrast to the On/Off controller, the thermostatic control using `regStep()` results in smooth curves for mass flow and temperature. However, the models differ greatly in their simulation performance: although the effort to simulate one step is the same (equal number of variables, continuous time states and equation systems), the On/Off controller solution increases the simulation time by a factor of 2.3. The reason is that it generates events with every switching (two per hour on average), and that after each event, the solver (Dassl, tolerance $1 \times 10^{-4}$) restarts with very small simulation time steps of about 1 s. Thus, the solver calculates, on average, 39 steps per hour instead of 18 with `regStep()` implementation.

### 2.2.3. Limiting Dynamics of the Models

Physical processes within a DHN occur on different time scales. While a water hammer runs at the speed of sound through the pipes (about 1000 m/s), thermal energy transport is bound to the flow velocities in the pipes (about 1 m/s). Furthermore, actuators (e.g., valves and pumps) and the corresponding control loops react within seconds to minutes to changing conditions, whereas heat load profiles for annual simulations are typically available at hourly resolution. These examples illustrate that the dynamic processes in real DHNs take place at very different orders of magnitude.

Thus, when modeling DHNs, it is useful to define the time scale of dynamic effects that is within scope as well as that which is not. Then, the dynamics of the models can be restricted to this time scale so that the effects out of scope are not modeled and simulated to avoid additional computational effort. Jorissen et al. [20] mention that especially reducing the fastest dynamics in the system may be beneficial, but they also warn that this measure may be non-physical. However, preventing the HC model from imposing instant changes of mass flows is not a limitation of the model but a realistic feature that represents opening or closing time of the valve that regulates the mass flow.

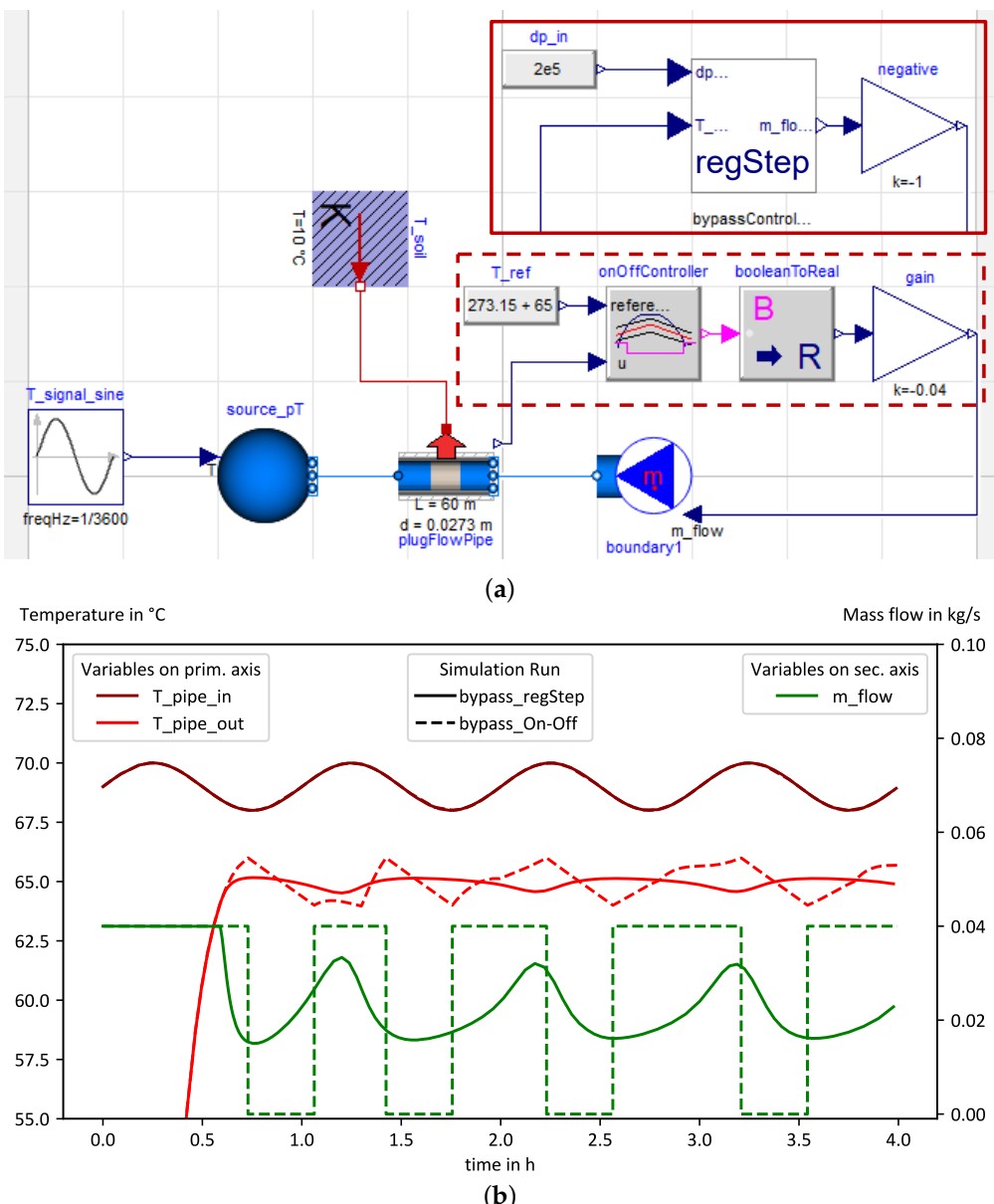

(**a**)

(**b**)

**Figure 2.** Simple model of bypass control to demonstrate the effectiveness of `regStep()` (solid rectangle and lines) in comparison to an On/Off controller (dashed rectangle and lines). (**a**) Diagram layer: A plug-flow pipe model instance is fed with a varying inlet temperature. The bypass control models (within the rectangles, both variants are shown) ensures a certain outlet temperature by providing the set-point for the mass flow source at the pipe outlet; (**b**) Simulation results: The green lines indicate mass flows, the dark red line is the pipe inlet temperature and the red lines are the pipe outlet temperatures.

With this in mind, it is obvious that modeling water hammers is out of the question for annual simulations of DHNs. It requires the use of a compressible fluid model with a dynamic mass balance in the pipes and entails the calculation of effects that happen within fractions of a second. But even without this, an appropriate tuning of the dynamics is important, as can be seen from the CPU time of the proposed HC model with two different time constants for mass flow (*main* and *fast*) in Table 2 in Section 4.2.7. The only difference between the two models is the different values for the time constant of the mass flow in the HC model instances. The faster dynamics result in an increase in CPU time by 17%, because about that many more steps have to be calculated.

## 3. Description of the Proposed Heat Consumer Model

The implementation of the proposed HC model follows the previous considerations to keep low the computational effort for the simulation of the DHN.

### 3.1. Heat Consumer Model Design

Figure 3 offers an overview on the design of the proposed HC model. The open-loop design (more details in Section 2.2.1) is obvious as there is no fluid model connecting the supply line with the return line. This separates the large equation system for mass flows and pressures in the pipe network into two smaller equation systems that require less computational effort. The model contains control blocks for load and optional bypass that calculate set-points for mass flow (and return temperature in the former case), and a block that combines the two flows and two mass flow sources that generate the prescribed in- and outflow. The calculation of the load mass flow is based on the input load time series (connected via a data bus), the measured differential pressure in the load and an input value of the supply line temperature which is connected to the end of the supply line pipe model right before the HC to provide a valid temperature value during zero-flow periods. The differential pressure signal is also connected to the data bus for further processing by the network's differential pressure control.

This design contains a minimal number of components with fluid models (only the two mass flow sources), which reduces the effort required to compute medium properties to a minimum. The used mass flow sources are very simple and lightweight models that neither contain a pressure loss calculation nor introduce state variables to the model. Furthermore, this design does not require any control loops that may oscillate and reduce the simulation time step.

Basically, this design boils HC behavior down to its core: setting the mass flow and extracting heat from it if possible. Additional features ensure realistic behavior: the mass flow calculation introduces a time constant as a simple representation of reaction times of controllers and regulators in a real HC. Furthermore, the HC model takes limits of mass flow with respect to differential pressure into account, without calculating pressure drops in its components. Thus, this simple design is able to reproduce the behavior of a HC with correctly dimensioned components and properly functioning control and regulation devices. A detailed explanation of these features is provided in the following section.

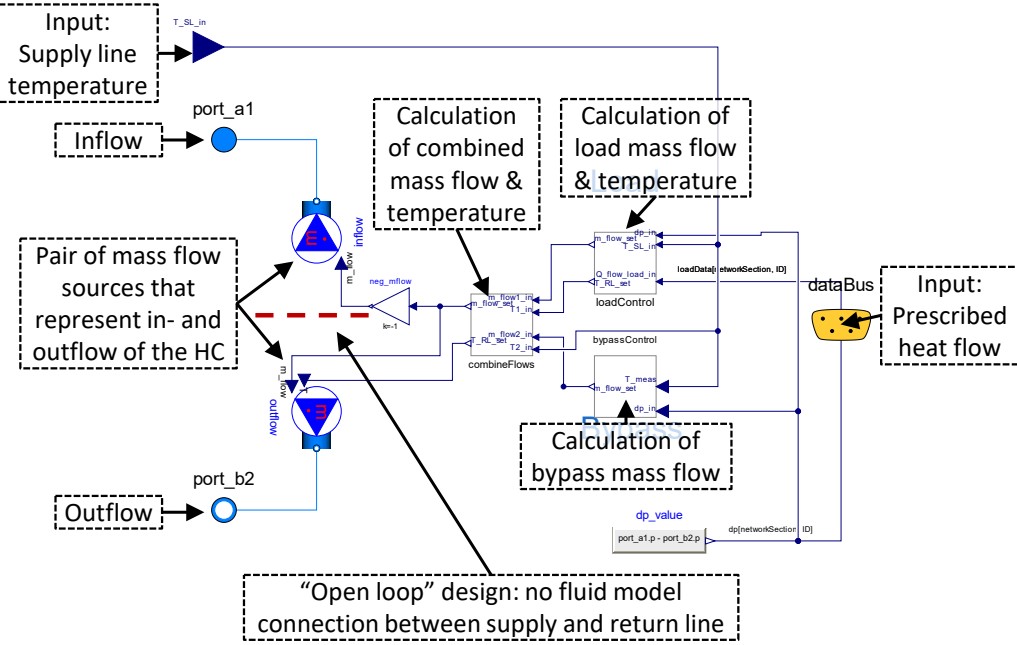

**Figure 3.** Diagram layer of the proposed HC model.

### 3.2. Determining Load Mass Flow

The determination of load mass flow $\dot{m}_{\text{load}}$ within the load control block deserves special attention. It is calculated from prescribed heat flow $\dot{Q}_{\text{load}}$, heat capacity of water $c_p$ and the temperatures in supply $T_{\text{SL}}$ and return line $T_{\text{RL}}$ according to Equation (1).

$$\dot{m}_{\text{load}} = \frac{\dot{Q}_{\text{load}}}{c_p \cdot (T_{\text{SL}} - T_{\text{RL}})} \tag{1}$$

However, a robust implementation of this simple equation according to the previously described goals and strategies requires some more details.

First, there may be situations when the supply line temperature is close to or even below the set-point return line temperature, causing Equation (1) to yield infinite or negative values. In such cases, it can be assumed that the set-point temperature for the supply line of the secondary side of the substation is not reached, causing the controller and regulator of the substation to increase the primary mass flow as much as possible. Furthermore, it is assumed that in these situations, the amount of heat extracted from the mass flow is negligibly small. The increase in primary mass flow is modeled as a smooth transition between normal and undersupply operation with a `regStep()` formula, increasing the mass flow from $\dot{m}_{\text{load}}$ (calculated according to Equation (1)) to $\dot{m}_{\text{max}}$ (see next paragraph for details). The prescribed return line temperature changes to the actual supply line temperature when the difference between flow and return line temperature crosses zero using a `smooth()` operator to avoid events.

Second, $\dot{m}_{\text{load}}$ is limited to a meaningful range between zero and maximum mass flow $\dot{m}_{\text{max}}$. As the available differential pressure $dp$ determines the maximum possible mass flow through the heat exchanger of the HC, $\dot{m}_{\text{max}}$ is not constant but calculated from differential pressure $dp$, minimal required differential pressure $dp_{\text{min}}$, mass flow at nominal load conditions $m_{\text{nom}}$ and factor $f_{\text{mflow,max}}$ that accounts for oversizing of the components according to Equation (2). A square-root approximation is chosen to implement the general dependence of mass flow on available pressure difference.

$$\dot{m}_{\text{max}} = f_{\text{mflow,max}} \dot{m}_{\text{nom}} \sqrt{\frac{dp}{dp_{\text{min}}}} \tag{2}$$

This feature requires the HC model to be used within a DHN model with a differential pressure control that assures minimum differential pressure $dp_{\text{min}}$. In cases where the heat supply unit is not able to provide sufficient differential pressure at HCs, the model reduces $\dot{m}_{\text{max}}$, so at a certain point, the actual mass flow $\dot{m}_{\text{load}}$ is limited and finally reaches zero when the differential pressure is zero or below. This simple and computationally light implementation allows for detection of such pressure undersupply situations and provides insight into the data regarding the affected units and the extent of influence. This feature might add an algebraic loop to the model, as it introduces an interdependence of differential pressure and mass flows at the HCs. The resulting nonlinear equation system can be solved during each time step at high computational effort. This is avoided by the next feature.

Third, in line with Section 2.2.3, the mass flow variable has time constant `tau_m_flow`. This is implemented by introducing two mass flow variables: `m_flow_fast` is calculated according to Equation (1), while the mass flow to be set in the model, `m_flow_set`, is delayed by using time constant `tau_m_flow` as shown in Listing 1 (implementation adapted from Lawrence Berkeley National Laboratory ([36], Section 3.3.4)). This feature introduces state variables into the mass flow calculation so the algebraic loop mentioned in the previous paragraph is avoided.

**Listing 1.** Implementation of the mass flow time constant.

```
der(m_flow_set) = (m_flow_fast-m_flow_set) / tau_m_flow;
```

Fourth, as an optional feature, the consumer model is able to include a hysteresis: whenever the prescribed heat flow falls below the switch-off threshold, $\dot{m}_{\text{load}}$ is set to zero until the value rises again above the switch-on threshold. This feature may reduce computational effort if the time series of the prescribed load value includes longer periods of negligibly low values. Instead of simulating them in detail, they are omitted. However, this feature triggers events whenever the thresholds are crossed at the cost of additional computational effort, so the simulation time may even increase.

Finally, again, optionally, the model may keep track of the deviation between prescribed and actual heat flow that may occur due to the time constant, hysteresis and undersupply situations via additional variable `loadDiff`. This deviation is accumulated and then used to correct the heat flow signal so that the missing or surplus heat becomes balanced. The implementation of the load correction is shown in Listing 2. To account for the inertia of the involved thermal and technical processes (e.g., cooling of a room takes some time, thermostatic valves of radiators react smoothly to deviations from the set point), the accumulated deviation is added to the load value over a certain period of time (`fac_tau_corrLoad*tau_m_flow`, here set to 15 min). Furthermore, the corrected heat flow is limited within a meaningful range from zero to a maximum value, here 1.5 times the nominal heat load.

**Listing 2.** Implementation correction of load deviations.

```
der ( accLoadDiff ) = loadDiff ;
Q_flow_load_corr = Q_flow_load_internal -
                   accLoadDiff / ( fac_tau_corrLoad * tau_m_flow );
```

### 3.3. Determining Bypass Mass Flow

The bypass is intended to maintain a certain minimum temperature in the supply line before the HC. To that end, the bypass control sets the bypass mass flow depending on this temperature. When it is high enough, the mass flow is zero. Once the temperature approaches the set-point temperature that the bypass maintains, the mass flow is gradually increased within a bandwidth around the set-point temperature until it finally remains at a maximum mass flow if the supply line temperature is at or below the lower end of the bandwidth. Once again, this behavior is implemented using `regStep()`, as this yields a smooth characteristic and does not trigger events.

In addition, maximum available bypass mass flow $\dot{m}_{\text{bypass,max}}$ is reduced below its nominal value $\dot{m}_{\text{bypass,nom}}$ if differential pressure $dp$ is below minimum required value $dp_{\text{min}}$ following Equation (3)).

$$\dot{m}_{\text{bypass,max}} = \dot{m}_{\text{bypass,nom}} \cdot \min\left(1; \sqrt{\frac{dp}{dp_{\text{min}}}}\right) \tag{3}$$

The return temperature of the bypass mass flow is simply set to the actual supply line temperature, as it is assumed that no heat is extracted from this mass flow.

## 4. Evaluation of the Heat Consumer Model

To evaluate the HC model concerning the results and its effects on simulation performance, a demonstration network is modeled and simulated in Dymola using the models of the in-house *DHNSim* library. The simulations are run with the same network and heat load data for different HC model configurations to investigate their effects on simulation results and performance. Additionally, simulations are also performed with the two open-loop demand models from `AixLib.Fluid.DistrictHeatingCooling` (`VarTSupplyDp` and `VarTSupplyDpBypass`, constant return temperature) and the most simple configuration of `DisHeatLib.Demand.Demand` (constant return temperature, linearized flow characteristic in the flow unit). For the latter, it was a difficult task to obtain stable operation of the HCs due

to oscillations in the internal control loops. Table 1 shows an overview on the simulation runs and their specifications.

**Table 1.** Overview of the simulation runs.

| Name | Specifications (HC Model and Other) |
| --- | --- |
| *main* | *DHNSim*, constant return temp., no load correction, `tau_m_flow = 180 s`, with bypass, no hysteresis |
| *corrLoad* | alike *main*, but with correction of deviations between prescribed and actual load value |
| *fastDynamics* | alike *main*, but `tau_m_flow = 30 s` |
| *noBypass* | alike *main*, but no bypasses |
| *hysteresis* | alike *main*, but with hysteresis to swith load mass flow off |
| *onePipe* | alike *main*, but pipe network contains only one pipe |
| *AixLib* | *AixLib* open-loop demand model, constant return temperature |
| *AixLibBypass* | *AixLib* open-loop demand model, constant return temperature, with bypass |
| *DisHeatLib* | *DisHeatLib* demand model, constant return temperature, linearized flow characteristic |

### 4.1. Demonstration Network

The basic demonstration network is a fictional, simple DHN with one supply unit and six HCs. The pipe network consists of 17 pairs of pipes (supply and return line), including house lead-in pipes, and contains one loop to introduce a certain degree of complexity (the loop results in a non-linear system of equations for the mass flows and pressures). The pipe closing the loop (DN 50) is split into two parts to obtain a temperature value in the middle of the pipe for analysis. To analyze the effect of different network sizes, this basic layout ("small") is repeated three times ("medium") and nine times ("large"), branching off after the first network pipe. The layout and the main parameters are depicted in Figure 4.

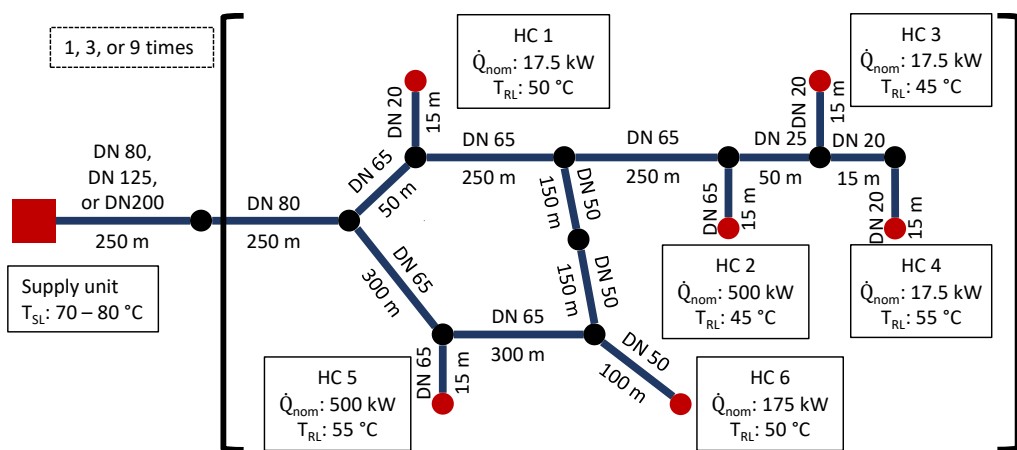

**Figure 4.** Layout of the demonstration network with the main parameters. For HCs (red circles, HC 1 to HC 6) nominal heat load and constant return line temperature, at the supply unit (red square) the supply line temperature and for pipe segments (blue lines) nominal diameter and length are given. Black circles mark pipe sections without heat extraction. Simulations are run for different network sizes where the network part in brackets exists 1, 3 or 9 times after the first pipe.

The HCs are simulated with six real, measured heat load profiles from an existing DHN with a resolution of 15 min. For the "medium" and "large" simulations, the profiles are reused with random variations (normal distribution, standard deviation 10%) so that the peaks and valleys do not perfectly coincide. The heat load profiles consist of exemplary periods for high load, medium load, low load (each three days) and an undersupply situation (two days with a temporary drop of supply line temperature to 50 °C). The supply line temperature at the supply unit is set via a temperature curve between 70 and 80 °C, apart from the undersupply situation, where the actual measured supply temperatures

are used. During the simulation, these values are interpolated using smooth splines with `Modelica.Blocks.Sources.CombiTimeTable`.

The heat load profiles, being real measurement data, show higher dynamics (frequent peaks and valleys) than common synthetic heat load profiles. This is most pronounced during the medium- and low-load periods at HC 5 and HC 6, which show frequent switching between zero and substantial load values. Furthermore, most of the load profiles include periods with zero load for some hours. Thus, these heat load profiles are challenging yet realistic examples of heat load profiles that may be used in the simulation of DHNs.

### 4.2. Evaluation of Simulation Results and Performance

To check whether the demonstration network is configured realistically, some general indicators are estimated from the results of the *main* simulation run (low load = winter, medium load = spring and autumn, high load = summer). For the basic small demonstration network and the *main* simulation run, the estimation yields a total annual heat demand of 3.2 GWh/a, relative heat losses of 12% and a relative hydraulic energy for the circulation of 0.22%. Given the route length of 2.2 km, the linear heat density is 1.3 MWh/(m· a). The mass flow weighted mean temperatures at the supply unit are 72 °C in the supply line and 48 °C in the return line. These values are considered plausible for a medium-sized DHN with network temperatures as low as possible while still supplying old buildings and preparation of domestic hot water.

#### 4.2.1. Comparison of General Simulation Results

In general, the simulation results of the different HC models should be similar. In the following, the results are compared to the *main* simulation run and major differences are reported and explained.

The total heat from the supply unit does not differ by more than 3% compared to the *main* result for all models and periods, which indicates a good agreement of the models.

During the low-load period, it makes a major difference whether the HC model includes a bypass. Compared to *main*, models without bypass (*noBypass*, *AixLib* and *DisHeatLib*) result in about 9% less heat losses, because the network is not kept hot and lower return temperatures occur. Furthermore, they yield a 20 to 30% higher maximum heat flow due to mass flow peaks after the supply line temperature cools down. In addition, the maximum pressure difference at the supply unit is 13 to 19% lower due to lesser mass flow in the network. Accordingly, the total hydraulic energy at supply unit is about 30% lesser than with bypasses in this period.

Furthermore, during the low-load period, *AixLibBypass* has 7% less heat losses than *main*, as the constant bypass flows are not sufficient to keep the network hot (but also should not be tuned to the necessary value, because too much load would be omitted then).

In the undersupply period, the *AixLib* models have 13% lower maximum pressure differences, as they assume a constant minimum temperature difference (set to 5 K in this case), while *DHNSim* models set mass flows to a maximum allowed value. Furthermore, the hydraulic energy is 20 to 30% lesser without bypasses (*noBypass*, *DisHeatLib*) and 56% lesser for the *AixLib* models due to lower mass flows in both cases.

Another difference is that the maximum differential pressure at the supply unit is 19% higher for the *AixLib* models during the high-load period due to a single, probably faulty, data point in the heat load profile of HC 4 (critical path), with a prescribed heat flow of 35 kW (although rated to 17.5 kW). The *DHNSim* models limit the mass flow according to Equation (2) (here with $f_{\mathrm{mflow,max}} = 1.5$), which limits the heat load in this case to 18 kW.

#### 4.2.2. Effect of Mass Flow Time Constant and Load Correction

The comparison of the heat and mass flows at HC 6 for simulation runs *main*, *fastDynamics* and *corrLoad* in Figure 5 demonstrates that a heat load peak and bypass operation changes due to different values of the time constant and the optional correction of deviations of the actual heat load from the prescribed value.

The time constant delays and slightly reduces the heat load peak compared to the input signal due to the added dynamics. The smaller the value of `tau_m_flow`, the more immediate the reaction of the HC model to the input signal. Depending on the goals and available input data, the user of the model chooses a sensible value for `tau_m_flow`. For input time series at a resolution of 15 min to 1 h, a value of 180 s has proven to be suitable in previous simulation studies.

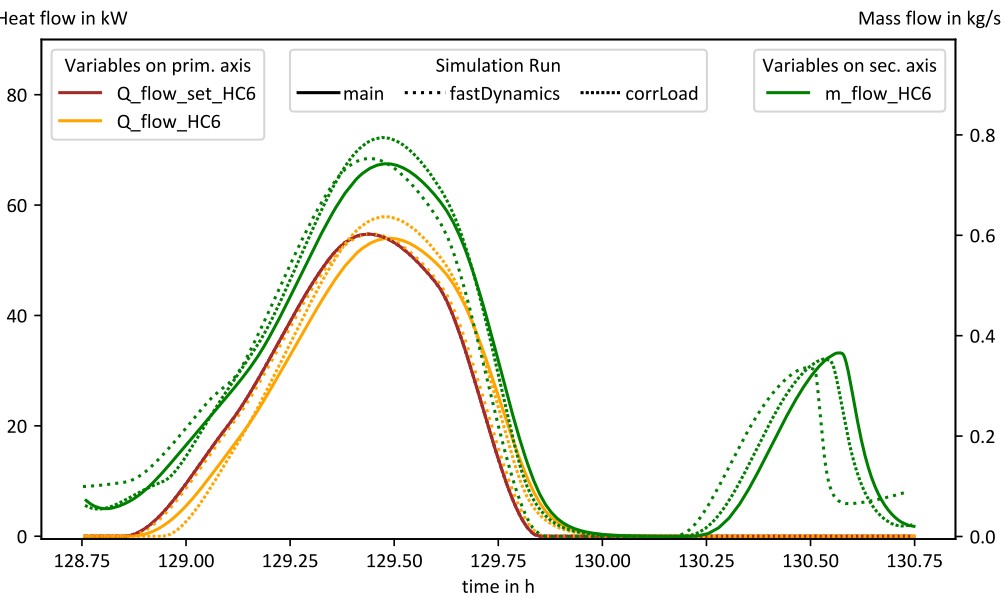

**Figure 5.** Example of the effect of the mass flow time constant and the load correction at HC 6. The brown line is the set-point heat flow, orange are the heat flows from the simulation results and green are the mass flows. The line styles indicate the different simulation runs.

In contrast, with load correction, the heat load peak is increased, because during the rising slope the actual heat load is below the prescribed value, so the model increases the heat flow over the prescribed value to equalize this deficit. During the falling slope, the heat load is reduced faster with load correction than without once the previous deficit is equalized.

Figure 5 also shows that `tau_m_flow` has an impact on bypass operation. After 130 h, the heat flow signal, and subsequently that of the mass flow, drops to zero. However, after a short zero-flow period, the supply line temperature (not shown for clarity) drops below the set point of the bypass, causing it to increase the mass flow. The bypass in *fastDynamics* reacts first because the zero flow starts earlier so the cooled house lead-in pipe becomes flushed earlier than in *main*. The slower dynamics in *main* finally cause a slightly higher peak mass flow, because the bypass mass flow depends on the temperature reaching the HC, which is lower the longer the water cools down. However, both configurations maintain the required supply line temperature at the HC. The bypass in *corrLoad* behaves similarly to *main*.

The other evaluated HC models show different behavior concerning dynamics as shown in Figure 6. In general, course and magnitude of the mass flow peak are similar to those of the *main* model. The *AixLib* model reacts immediately to the prescribed heat flow signal and follows it strictly during the heat flow peak. This leads to an immediate and steep rise of mass flow with a minor intermediate peak which results from the fact that once the house lead-in pipe is flushed, the supply temperature rises and the mass flow can be reduced to meet the heat load. In contrast, the *DisHeatLib* model shows a delayed answer, but then an even steeper rise of mass flow. The intermediate mass flow peak occurs a little later but similarly to *AixLib*. On the falling slope, the *DisHeatLib* model slowly approaches (but never reaches) zero mass flow. This behavior results from the model implementation with a control loop for the load mass flow.

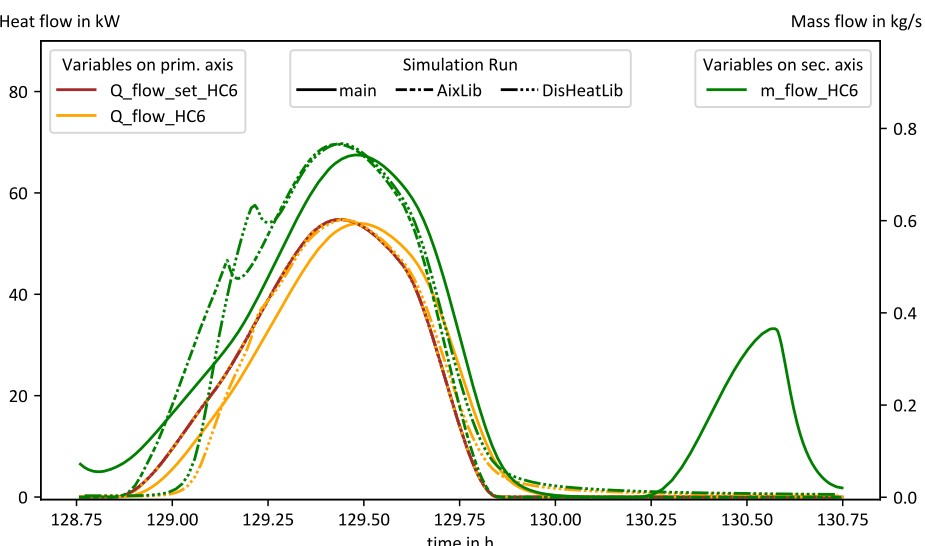

**Figure 6.** Comparison of dynamic behavior of the proposed HC model to other models. The brown line is the set-point heat flow, orange are the heat flows from the simulation results and green are the mass flows. The line styles indicate the different simulation runs.

### 4.2.3. Bypass Behavior

Bypasses are intended to maintain a small mass flow through HCs during zero-load periods so that the supply line temperature does not drop too much. Figure 7 shows the results for temperatures and mass flows at HC 6 during a period without heat load for *main* and *noBypass*. In *main*, the bypass starts to operate once the supply line temperature approaches 65 °C. The mass flow shows an decreasing oscillation, which is caused by the interplay of the thermostatic control approach and the delay due to the dwell time of the water in the house lead-in pipe. As long as the bypass operates and the heat load is zero, the return line temperature equals the supply line temperature. The bypass successfully maintains the required temperature of about 65 °C. Once the heat load rises (at 133.5 h), the return temperature smoothly drops.

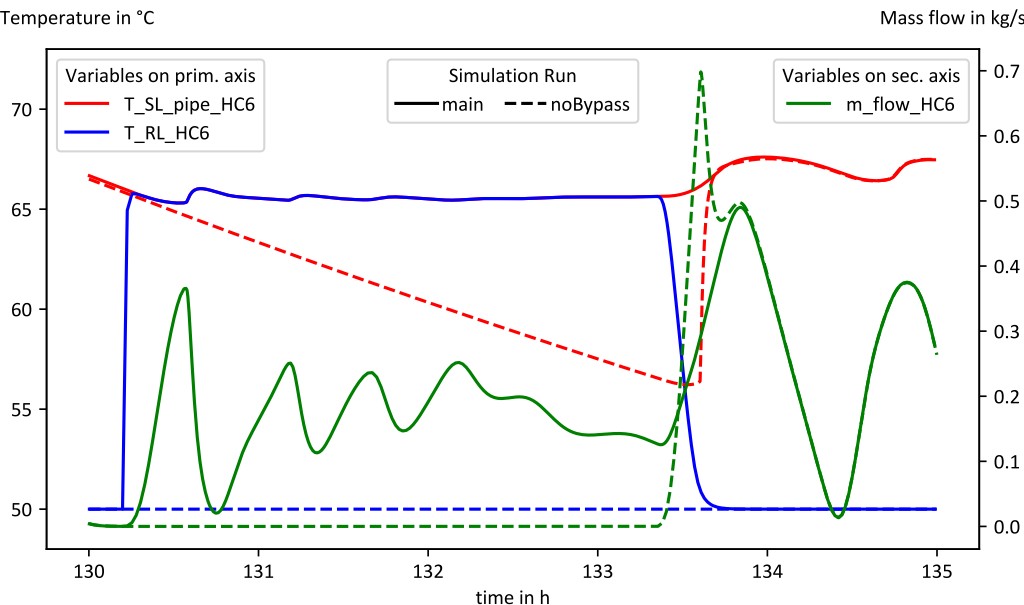

**Figure 7.** Demonstration of the bypass part of the HC model. The red lines are supply line temperatures, blue are the return line temperatures and green are the mass flows at HC 6. The line styles indicate the different simulation runs.

In contrast, in *noBypass*, the mass flow is zero during the period without heat load and the supply line temperature continuously drops. As a consequence, a mass flow peak occurs afterwards until the supply line temperature rises, causing steep slopes of mass flow and temperature. Nevertheless, the implementation of the HC is robust also without bypass due to the limited dynamics and maximum value of mass flow and its reduction if the differential pressure is too low. This prevents the HC model from imposing excessively high mass flows after zero-flow periods which might cause simulation failure.

For the demonstration network, *noBypass* implementation requires substantially less time to compute (see Section 4.2.7), which indicates that the reduced effort (no calculation of bypass mass flow) outweighs the computational effort to simulate the higher dynamics after zero-flow periods. In the end, it is up to the user whether a bypass is included, depending on whether it is intended and realistic to have it.

In general, the proposed bypasses work as intended. In the *main* simulation run, only HC 6 has supply line temperatures below 64 °C in the three-day low-load period, totaling 1 h, affecting a heat consumption of 9 kWh. In contrast, without a bypass, at all HCs supply line temperatures below 64 °C occur, with the strongest effect at HC 6 during the low-load period for 30 h and 300 kWh.

Figure 8 shows a comparison of the simulation results from *main* to the other HC models. In general, a major difference between models with and without bypass can be observed.

*AixLibBypass* maintains a constant minimum mass flow. If tuned properly, this approach succeeds in maintaining a sufficient supply line temperature. However, the bypass is active irrespective of the supply line temperature, whenever the load mass flow reaches the set point, as can be seen at 134.5 h. The implementation of the return line temperature is not robust (switches at an undetermined time instant, here 131.7 h) and causes abrupt changes. The bypass implementation of *AixLibBypass* does not reduce the duration of temperature undersupply significantly for two reasons. First, bypass operation does not depend on temperature but on heat load, so that in some periods the bypass does not act, although the supply line temperature is low. Second, and more importantly, it is not possible to tune the bypasses of critical consumers to a value that always maintains the supply line temperature, because it is chosen to limit the maximum allowed bypass mass flow to 10 % of nominal mass flow, as too much heat load is omitted otherwise.

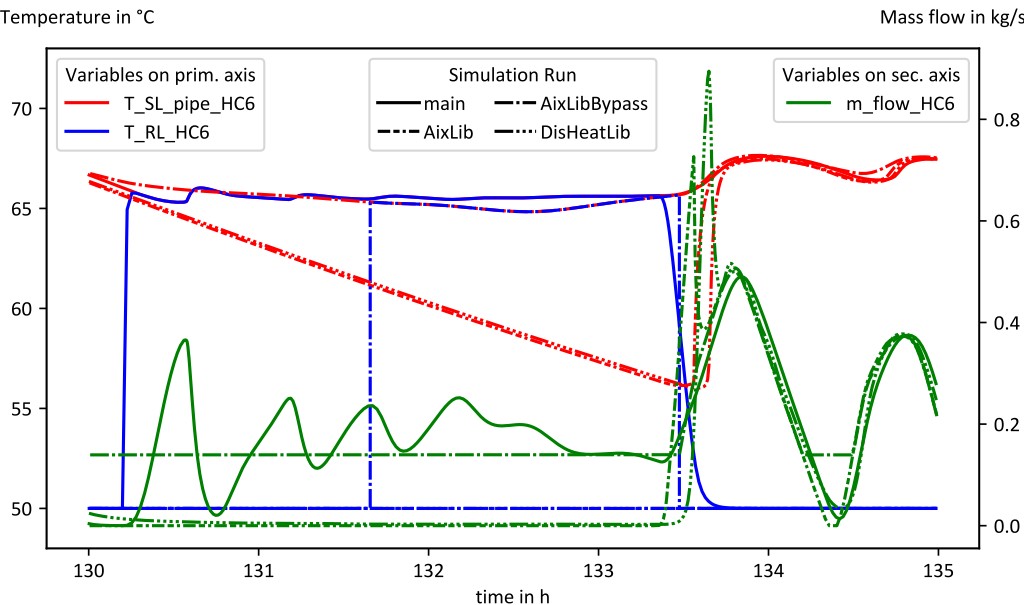

**Figure 8.** Comparison of the proposed HC model to other HC models for a period with bypass activity. The red lines are supply line temperatures, blue are the return line temperatures and green are the mass flows at HC 6. The line styles indicate the different simulation runs.

The two other models without bypass, *AixLib* and *DisHeatLib*, produce results that are very similar to those from *noBypass* (shown in Figure 7). During the zero-flow period, the supply line temperature constantly drops. Once the heat load is above zero, the mass flow rises. Both models show a strong and short peak of mass flow, which results from the cooled water in the house lead-in pipe. However, as *DisHeatLib* reacts with a certain delay, its mass flow peak is later and higher compared to *AixLib*.

### 4.2.4. Load Hysteresis

The hysteresis feature explained in Section 3.2 affects the behavior of the HC model when the prescribed load value is close to zero. Figure 9 shows an example for HC 3, comparing the results for heat and mass flows from *main* and *hysteresis*. In the period, two very low heat load peaks occur. The first (210.5–212 h) reaches values above the hysteresis thresholds. While the *main* result shows a smooth rise of heat flow following the set-point, *hysteresis* has zero heat flow until the threshold is reached (right before 211 h), followed by a steep rise of heat flow until the required values is reached. At the falling slope, the heat flow suddenly falls to zero once the switch-off threshold of the hysteresis is crossed (at 211.7 h). The second heat load peak (212–213 h) never crosses the switch-on threshold, so it is completely ignored in *hysteresis*. The mass flows are very similar, as they are dominated by the bypass mass flow that is similar for both results.

This example shows that the hysteresis approach may avoid the calculation of negligible heat flows. However, it imposes additional computational effort due to the events that are triggered whenever thresholds are crossed and the steep slopes that occur right after every switching.

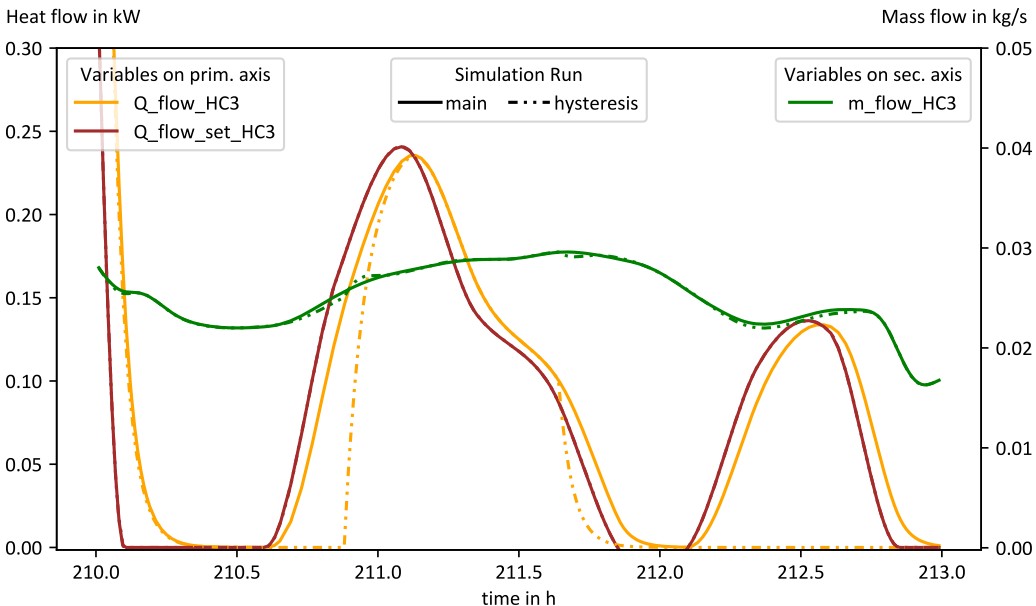

**Figure 9.** Demonstration of the effect of hysteresis at very small heat load peaks. The brown line is the set-point heat flow, orange are the heat flow results and green are the mass flows at HC 3. The line styles indicate the different simulation runs.

### 4.2.5. Undersupply

The proposed HC model is designed to provide plausible results during undersupply situations (excessively low supply temperature and/or differential pressure). Figure 10 shows the simulation results in such a period for *main* and *corrLoad* at HC 4, which is at the end of the critical path. The supply line temperature (upper graph) falls steadily and approaches the return line temperature, and as a consequence, the proposed HC model increases the mass flow (lower graph) to reach the needed heat flow.

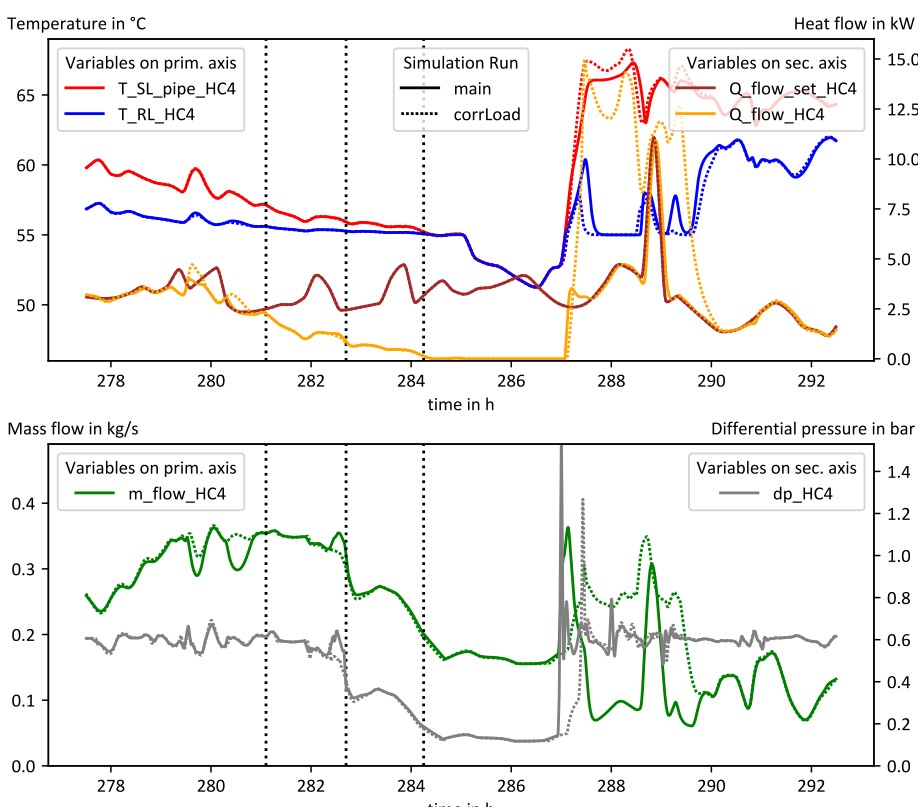

**Figure 10.** Demonstration of how the HC models deal with the undersupply situation (here for HC 4). The brown line is the set-point heat flow, orange are the heat flow results, red are the supply line temperatures entering the HC, blue are the return line temperatures, green are the mass flows and gray are the differential pressures between supply and return line at the HC. The line styles indicate the different simulation runs.

In *main*, the first phase of undersupply starts at about 281 h when the mass flow reaches its maximum (first vertical dotted line). From this point onward, the set-point heat flow is not covered. The second phase, starting after 282 h, is marked by insufficient differential pressure. Due to the enormous increase in mass flows in the network, pressure losses in the pipes rise, causing high differential pressures to be provided by the heat supply unit. At a certain point, the upper limit of differential pressure is reached so that the required minimum differential pressure at the HC (here 0.6 bar) is no longer maintained. As a consequence, the load model reduces the mass flow. Finally, after 284 h (third dotted line), the supply line temperature even drops below the set-point return temperature, so the heat flow is zero and the return line temperature equals the supply line temperature. Once the supply line temperature rises substantially at 287 h, the required differential pressure is restored and the HC returns to normal operation.

The model configuration *corrLoad* behaves similarly to *main* during the undersupply period (its densely dotted line is hidden behind the solid line of *main*). However, right after reestablishing a sufficient supply line temperature, the load correction comes into action causing a major load peak of 14 kW which lasts about 3 h until the heat load that was not covered is balanced. The load correction works properly for all loads, with deviations between set point and actual heat consumption below 0.1% for all HCs in the undersupply period. In contrast, other models result in substantial deviations of up to $-15\%$ for the proposed HC model without load correction and up to $-25\%$ for *DisHeatLib* (note that the actual undersupply lasts for only 6 hours in the two-day simulation period).

Figure 11 shows the simulation results from *main* in comparison to *AixLib* and *DisHeatLib*. *DisHeatLib* behaves similarly to *main*, as the flow unit in the model limits the mass flow to a maximum value according to the available pressure difference. The parameterization is

derived from nominal values and results in rather low maximum mass flows and more undersupply than *main*. In addition, oscillations of the mass flow control system can be seen, especially right after the undersupply period.

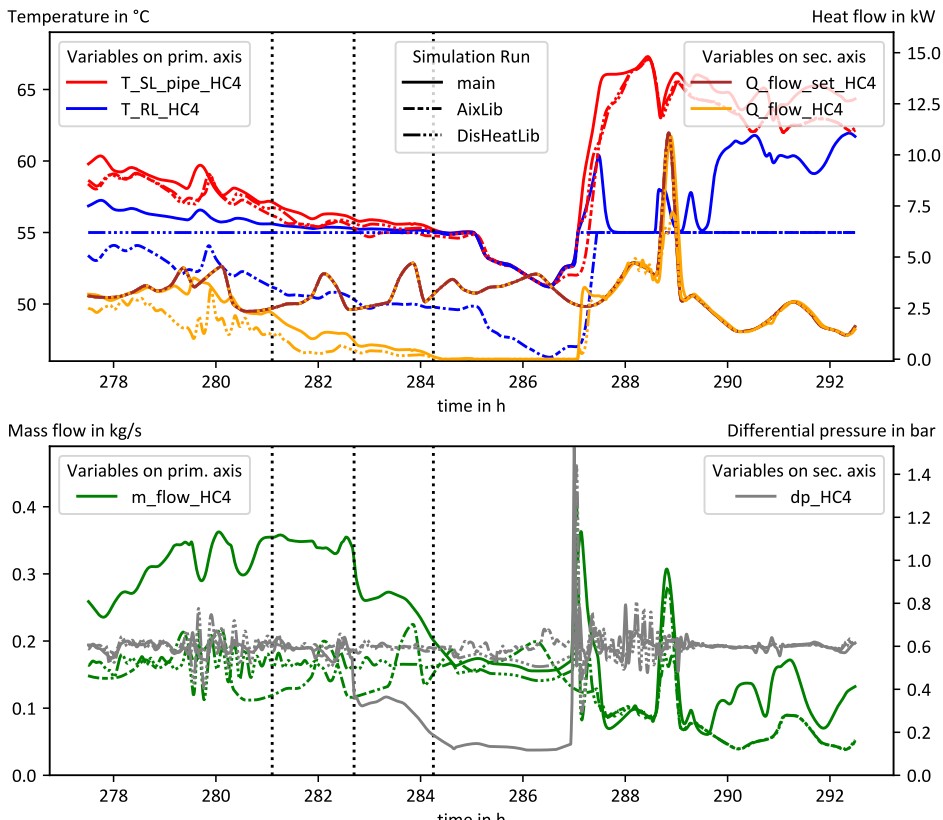

**Figure 11.** Comparison of the proposed HC model to other models in the undersupply situation (here at HC 4). The brown line is the set-point heat flow, orange are the heat flow results, red are the supply line temperatures entering the HC, blue are the return line temperatures, green are the mass flows and gray are the differential pressures between supply and return line at the HC. The line styles indicate the different simulation runs.

The *AixLib* model, however, deals with the situation differently. The model assumes a minimum temperature difference between flow and supply line (here 5 K) that is used whenever the supply line temperature is low. This implementation leads to smaller mass flows compared to the proposed HC model and lets the model follow the set-point heat flow, so no undersupply occurs. However, the return temperature drops to 45 °C, and might even drop further, which is unrealistic if the secondary return temperature of the actual HC is higher than that.

### 4.2.6. Comparison with Measurement Data from an Existing DHN

To further evaluate the plausibility of the HC model, measurement data from an existing DHN were used. A major difference to the presented HC model is that the return temperatures are not constant but change over time. This is a major weakness of the proposed model and the HC models from other libraries, and this limits the comparability between measurement and simulation.

Unfortunately, the data are only available as instantaneous values every 15 min, so the analysis of processes below that time step is not possible. Thus, an evaluation of the time constant was not possible. However, it is known from practice that the actuators of valves have runtimes of about 30 s to a few minutes; therefore, a time constant in that range is plausible.

The hysteresis feature is not intended to be physical; it rather represents an idea to improve the simulation time with very little deviation from the original time series. Thus, a comparison to measurement data is not suitable.

Bypass behavior does not necessarily require an actual bypass component. The measurement data indicate that HCs with instantaneous hot water preparation act like thermostatic bypasses at zero load because the substation maintains a small mass flow to keep the domestic hot water heat exchanger hot so that hot water preparation can start immediately when water is tapped. Figure 12 shows an exemplary period. Note that the secondary *y*-axis ranges only from 0 to 0.25, so the small values for heat and mass flow are visible.

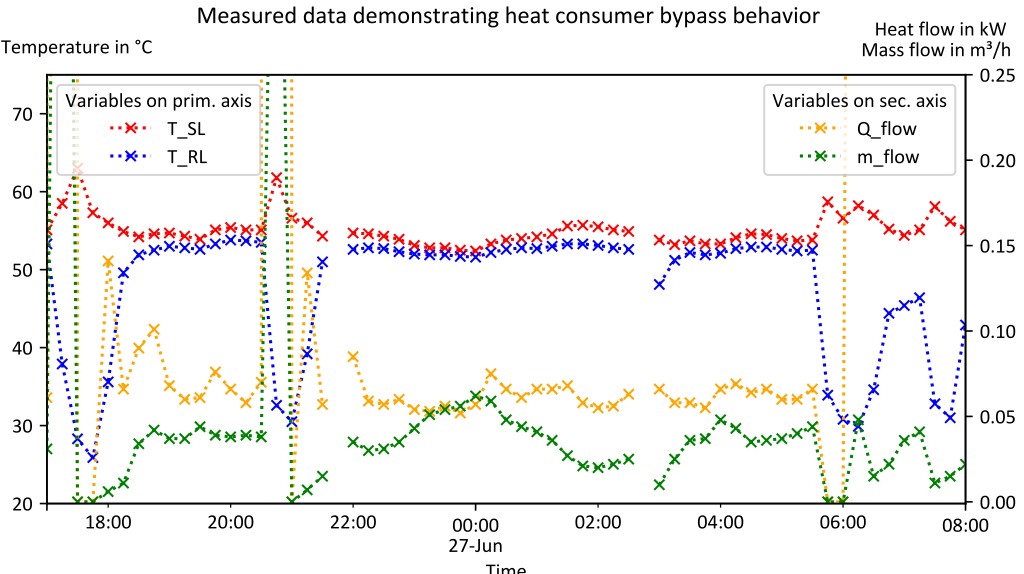

**Figure 12.** Measurement data from a HC with bypass behavior. Note that the data were only available as instantaneous values every 15 min (markers). The values in between are unknown. The dotted line is thus only intended to help track the course of each variable.

After a short heat load peak (corresponding to a tapping event) at 21:00 h, the mass flow and heat load are zero. Then, the supply line temperature drops and the mass flow starts to rise again. For the next 8 h, the substation shows the bypass behavior maintaining a temperature of about 55 °C, with a small mass flow, almost no cooling of the water and a negligible heat flow. The behavior is similar to a thermostatic valve, meaning that the more the supply line temperature drop, the greater the mass flow. This behavior shows great similarity to the results from *main* (see Figure 7).

For the undersupply behavior, appropriate measurement data are available to demonstrate that the behavior of the proposed HC model can be found in the real world. Figure 13 shows the measurement data of one HC that experiences the supply line temperature drop that was simulated as well. Due to the intermittent nature of this HCs heat load profile, the evaluation is only meaningful starting from 1:00 h in this case. From that moment on, it can be seen that the HC draws a high mass flow from the network but is not able to extract heat from it (supply and return line temperatures are almost equal). As the supply line temperature drops further, other HCs in the network increase their mass flow, so the differential pressure at this HC is reduced, and in consequence, the mass flow is reduced.

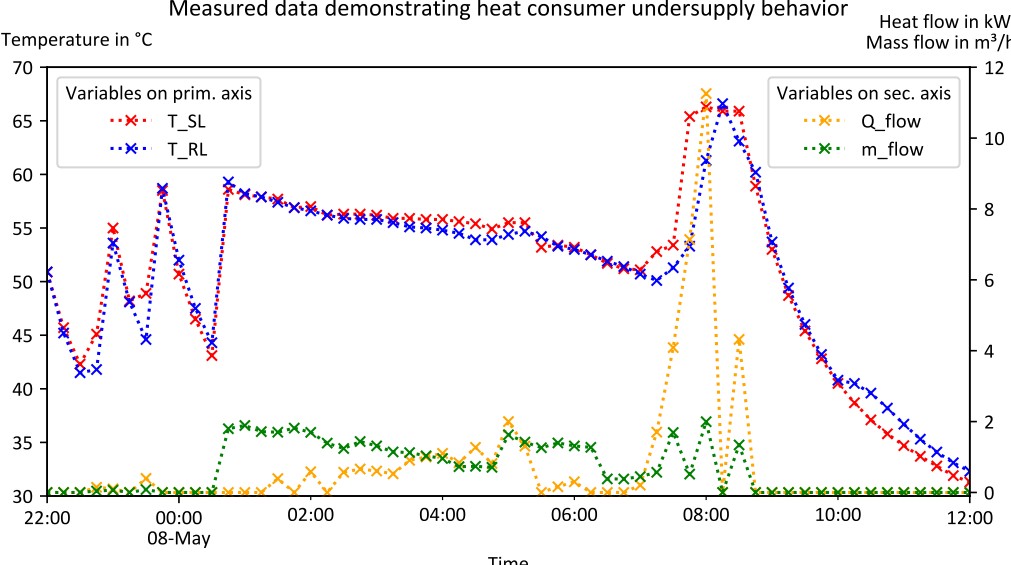

**Figure 13.** Measurement data from a HC in an undersupply situation. Note that the data were only available as instantaneous values every 15 min (markers). The values in between are unknown. The dotted line is thus only intended to help track the course of each variable.

In contrast to the simulation, from 5:00 to 6:15, the mass flow is on a higher level again. This effect can be observed at all HCs that increased their mass flow drastically due to insufficient supply temperature. Most probably, the differential pressure in the network increased during that period by manually setting the pump speed to a maximum to try to overcome the undersupply situation.

Once the supply line temperature is restored to a sufficient value at 7:30, the HC has a short but massive heat load peak and draws a high mass flow until its heat demand is fulfilled. Then, its mass flow and heat load is zero again, so the water in the house lead-in pipe cools down.

All in all, the undersupply behavior is very similar to that of the proposed HC model with almost no extraction of heat, a high mass flow that drops as the differential pressure is reduced and a sharp heat load peak after the undersupply situation (see Figure 10).

### 4.2.7. Simulation Performance

For a profound analysis of the influence of the different implementations of HC models and network sizes on simulation performance, the CPU time for integration is evaluated (best of three runs, integration algorithm *Dassl*, tolerance $1 \times 10^{-4}$, on a machine with CPU Intel i5-4300U @ 4x1.9 GHz, RAM 8 GB).

Figure 14 shows the CPU time relative to the *main* model. *OnePipe* requires only a small fraction of CPU time compared to *main*, which proves that the HC model itself does not require much computational effort. However, the implementation of the HC model causes major variations in CPU time for the same network, with an increasing importance for larger models (more than a factor of five between *noBypass* and *corrLoad* for the large model). The models with open-loop design and without bypass (*noBypass* and *AixLib*) have the lowest and very similar CPU times. The proposed model *main* is the fastest with a bypass. *FastDynamics* lead to a minor increase in CPU time. *DisHeatLib* requires 33 to 40% and *AixLibBypass* 55 to 60% more computational effort. The implementation of *hysteresis* causes the long CPU times, especially for the large network. The use of the load correction in *corrLoad* leads to the longest CPU times (100 to 150% longer than *main*).

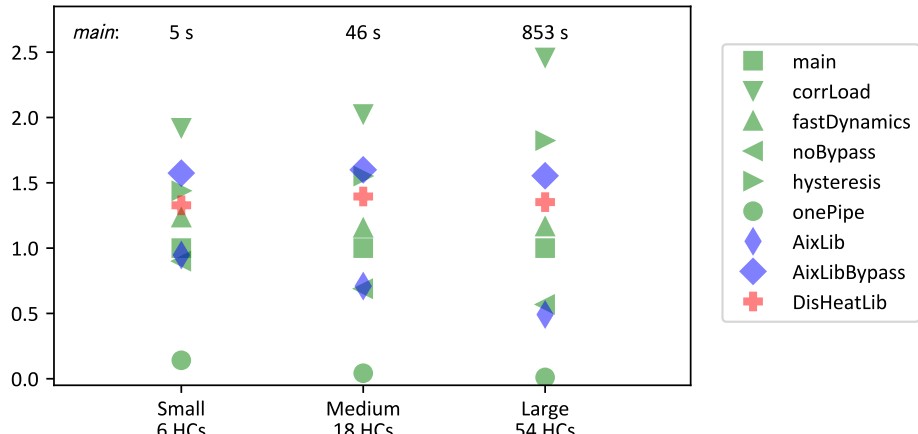

**Figure 14.** Comparison of CPU times of the different HC models at different network sizes. The green markers refer to different configurations of the proposed HC model, while the blue and red markers refer to HC models from other libraries (*AixLib* and *DisHeatLib*, respectively). Values are shown relative to *main*.

The absolute values (indicated in Figure 14 as well) show that CPU time scales non-linearly with model size, reaching about 850 s (*main*) for the large demonstration network. Assuming a linear dependence on simulated time, an annual simulation of the large network takes about 8 hours, which is acceptable but substantial, which stresses the importance of a careful design of the HC models.

Table 2 offers an overview on all simulation runs for the large DHN with their CPU time alongside selected model properties and simulation characteristics that help identify causes for differences in simulation performance.

**Table 2.** Results of CPU time alongside selected model properties and simulation characteristics for the large DHN.

| Simulation Run | CPU Time in s | Result Points | State Events | Jacobian Evaluations | States | Variables |
|---|---|---|---|---|---|---|
| *main* | 853 | 32,613 | 1851 | 8829 | 1000 | 15,396 |
| *corrLoad* | 2098 | 62,313 | 2108 | 23,511 | 1054 | 15,450 |
| *fastDynamics* | 994 | 37,678 | 2233 | 10,629 | 1000 | 15,396 |
| *noBypass* | 485 | 29,917 | 1622 | 8004 | 946 | 15,288 |
| *hysteresis* | 1555 | 54,027 | 9243 | 16,336 | 1000 | 15,396 |
| *AixLib* | 419 | 28,715 | 1654 | 7018 | 1000 | 15,612 |
| *AixLibBypass* | 1325 | 65,383 | 5371 | 24,769 | 1000 | 15,774 |
| *DisHeatLib* | 1153 | 34,784 | 1924 | 7682 | 1054 | 16,422 |

The results reveal that the number of result points has a direct and almost linear impact on CPU time. A simple linear fit of CPU time as a function of the number of result points yields a high coefficient of determination of 0.70. Figure 15 shows both measured and fitted CPU times for the large model. The deviations of the real measured CPU times from the fit may be caused by various factors that cannot be clearly identified due to the complexity of the computations. Some of these factors are the number of state events that occur, the number of Jacobian evaluations, the complexity of the resulting equation systems and the number of variables and states to be computed.

*AixLib* and the proposed HC model without bypass, *noBypass*, are much faster than expected from the fit and have a low number of state events and Jacobian evaluations. *DisHeatLib*, on the other hand, has a 30% higher CPU time measured than that according to the fit. As the model does not have an increased number of state events or Jacobian

evaluations, this increase can be attributed to the fact that this HC model does not follow the open-loop design which leads to solving complex systems of equations. Model *hysteresis* triggers by far the most events and thus has high CPU times. Similarly, *corrLoad* has a strongly increased CPU time that may be caused by the high number of Jacobian evaluations and a slightly increased number of states and variables to be calculated. Therefore, hysteresis and load correction should be used with care as these features may substantially increase the simulation time. Finally, *AixLibBypass* has a much shorter simulation time than expected from the fit, which cannot be attributed to any of the analyzed properties.

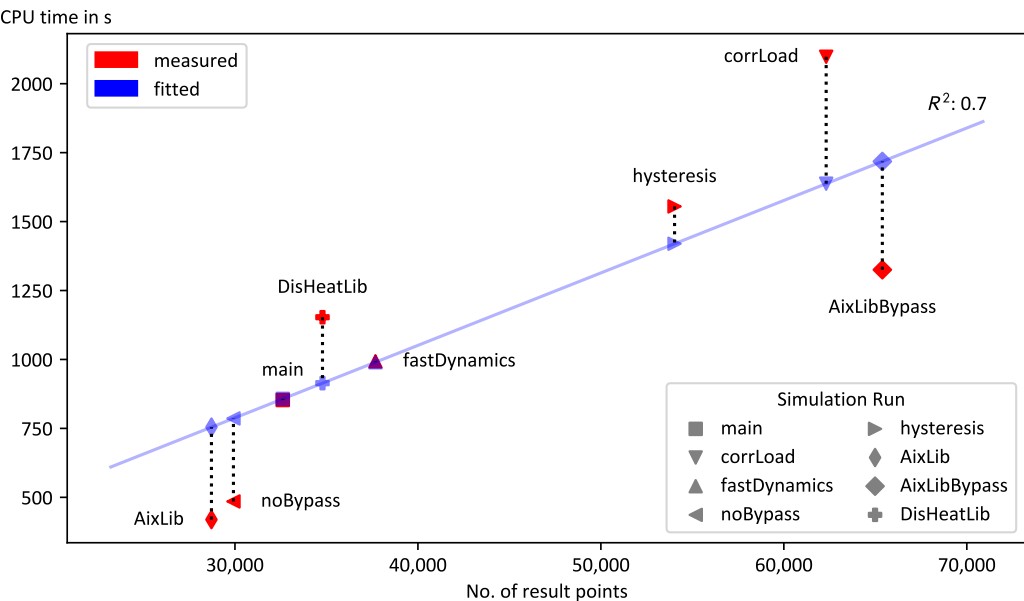

**Figure 15.** Dependence of CPU time on number of result points for the large model (54 HCs). Red corresponds to measured values and blue to the linear fit over the number of result points. The simulation runs are indicated with different symbols.

### 4.3. Limitations of the Proposed HC Model

Naturally, the simplified implementation of the HC model results in a number of limitations which are discussed in this section.

First, the simulation needs an external heat load profile which might either be derived from measured data or be generated artificially. However measurement data are only available for existing HCs and usually have limited quality (potential issues concerning accuracy, data availability for all HCs, gaps in the time series). In contrast, artificial heat load profiles can be generated with good accuracy for space heating loads, but they usually represent a smoothed profile for a whole district and ignore specifics of single HCs (e.g., dimensioning of components, optional night-time setback, user behavior). For domestic hot water, it is even more difficult to obtain suitable heat load profiles. Its characteristics are highly dependent on user behavior (tapping events) as well as the hydraulics (direct preparation or storage, circulation). Thus, providing valid heat load profiles is a tricky task which is left to the user of the proposed HC model.

Second, it may be a limitation that the model does not provide an option to change parameters during the simulation. In reality, set points in a controller, such as the temperature that a bypass maintains, may change during the course of the day or the year via schedules or at an unpredictable point of time, when users change these values.

Third, despite its ability to react plausibly to changing pressure and temperature conditions in the DHN, the HC model is simple and shows idealized behavior. An example is the undersupply model which assumes that the heat load is zero once the temperature in the supply line is as low as the prescribed return line temperature. This is not exactly true, as a HC still extracts heat to some extent during undersupply (see Figure 13). Thus,

the results from undersupply simulations somewhat differ from real scenarios. Nevertheless, we believe that the HC model is accurate enough concerning undersupply to draw general conclusions whether undersupply may occur and which HC would be affected to what extent.

Finally, and by far the most importantly, in our opinion, there exists limitation in the model since it uses a set point for the return temperature or the temperature difference (either constant or from an input time series) for heat extraction from the mass flow. Thus, the model is not able to reflect the dependence of the return temperature on the flow temperature or other influences such as current heat load value or type (space heating or domestic hot water) or the time of the day (e.g., important when the space heating controller includes a night-time setback). Providing an input time series for the return temperature is difficult, because either measured data must exist or detailed simulations of the building and its heat distribution system must be performed. Therefore, usually, users are left with using a constant return temperature, which is certainly not very realistic throughout the course of a year. Thus, there is a need for robust validated models for the return temperature depending on supply temperature, current heat load and other relevant factors, all of which are simple enough to be included in simplified HC models.

## 5. Conclusions

This article describes the development and evaluation of a simplified HC model using Modelica. The results demonstrate its capability to produce plausible results for the whole range of operating conditions, including undersupply and bypass operations. The comparison of the proposed model to models from other libraries proves that it is a significant improvement towards fast simulations of DHNs. The equivalent HC model from *AixLib* with bypass results in 55 to 60% longer CPU time and the most simple demand model from *DisHeatLib* (even without bypass) causes 33 to 40% longer CPU times. This improvement in simulation performance is achieved by a consequent simplified model design (including the open-loop approach), avoiding event generation and limiting the dynamics of the model.

Furthermore, the results prove that the implementation of the HC model has crucial impact on the computational effort of DHN simulations. Thus, developers of models for DHN simulations should carefully design and parameterize their HC models.

The users of the proposed model may choose whether their use case requires the implementation of a thermostatic bypass that maintains the supply line temperature and whether fast dynamics of the HC model are needed, as both options increase CPU time. Load hysteresis is not recommended for the used load profiles, as it triggers many state events, which causes a substantial increase in simulation time. The optional correction of deviations between actual and prescribed heat load should be used only if it is of high importance, e.g., for analysis of undersupply situations, as it drastically increases simulation time (up to 2.5 times in the analyzed example).

Future research on HC models should address a major weakness shared among many simplified models, including the proposed one: the assumption of a constant return temperature or temperature difference. A simple, robust and flexible model for the return temperature depending on the most important factors that reflects the main characteristics would be a major improvement. However, given the large number of system configurations and possible faults and malfunctions in district heating substations that influence the return temperature, this is an extremely difficult task.

The improved capabilities of the HC model enable further research using dynamic simulations of large DHNs. This includes analyses of hydraulic bottlenecks that take the reaction of HCs to pressure undersupply into account or investigations on minimum possible supply temperatures with respect to thermal undersupply and bypass flows.

**Author Contributions:** Conceptualization, J.Z., J.O., U.J. and K.V.; methodology, J.Z.; software, J.Z.; validation, J.Z. and J.O.; formal analysis, J.Z.; investigation, J.Z.; data curation, J.Z.; writing—original draft preparation, J.Z.; writing—review and editing, J.O., U.J. and K.V.; visualization, J.Z.; supervision, J.O., U.J. and K.V. All authors have read and agreed to the published version of the manuscript.

**Funding:** This research builds on results from research that was funded by the German Federal Ministry for Economic Affairs and Energy grant number 03SIN119.

**Data Availability Statement:** The data can be shared upon request, except for the original measurement data presented in Section 4.2.6 for data protection reasons.

**Conflicts of Interest:** The authors declare no conflicts of interest. The funders had no role in the design of the study; in the collection, analyses, or interpretation of data; in the writing of the manuscript; or in the decision to publish the results.

## Abbreviations

The following abbreviations are used in this manuscript:

| | |
|---|---|
| DHN | District heating network |
| HC | Heat consumers |

**Symbols**

The following symbols are used in this manuscript:

| Symbol | Explanation | Unit |
|---|---|---|
| $c_p$ | heat capacity of water at constant pressure | J/(kg K) |
| $dp$ | differential pressure | bar |
| $f$ | dimensionless factor | - |
| $\dot{m}$ | mass flow rate | kg/s |
| $\dot{Q}$ | heat flow rate | W |
| $T$ | temperature | °C |

**Subscripts**

The following subscripts are used in this manuscript:

| Subscript | Explanation |
|---|---|
| bypass | in the bypass part of the model |
| load | in the heat load part of the model |
| max | maximum allowed value |
| min | minimum required value |
| nom | nominal value |
| RL | in the return line |
| set | set point of the variable |
| SL | in the supply line |

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
