# Peer review of "Heat Consumer Model for Robust and Fast Simulations of District Heating Networks Using Modelica"

_electronics, doi:10.3390/electronics13071201_

Round 1

Reviewer 1 Report

Comments and Suggestions for Authors

Thank you for submitting your paper “Heat Consumer Model for Robust and Fast Simulations of District Heating Networks using Modelica” to the Electronics.

The paper focus on the modeling and simulation of District Heating Networks (DHN) using Modelica. The article discusses various aspects of DHN simulation, including the development of a Heat Consumer (HC) model designed for robust and fast simulations, strategies for efficient simulation, and evaluations of the model's performance.

The article is well written and well structured as a scientific text; the literature review is comprehensive and up to date. Here's a brief overview of suggestions for improvement across the main sections:

Introduction: Clarify the importance of the study in the context of current challenges in district heating. Emphasize the novelty of your approach compared to existing models and clearly state the research question or objective.

Simulation of DHN Using Modelica: Provide more background on Modelica's suitability for DHN simulations. A comparison with other modeling languages could help highlight its advantages.

Models for DHN and HC: For each library (AixLib, DisHeatLib, and DHNSim) discussed, a more critical evaluation of their strengths and weaknesses in the context of your study could be useful. Highlighting what your model improves upon or solves in these existing approaches would be valuable.

Strategies for Fast Simulations of DHN: This section could benefit from examples or case studies demonstrating the effectiveness of the proposed strategies in practical scenarios.

Description of the Proposed Heat Consumer Model: More detailed justification for the design choices and a clearer explanation of how they contribute to reducing computational effort while maintaining accuracy would strengthen this section.

Evaluation of the Heat Consumer Model: Incorporate a more detailed analysis of the results, potentially comparing them with real-world data or outcomes from other models to demonstrate the model's accuracy and efficiency.

Conclusions: Summarize the key findings more succinctly, emphasizing the practical implications of your research. Suggestions for future research based on your findings could provide a strong ending.

Comments on the Quality of English Language

good

Reviewer 2 Report

Comments and Suggestions for Authors

In this work, a model for simulations of district heating networks was developed, while the quality was verified. The topic fits the scope of the journal, and the paper title well describes the content of the work. I would like to congratulate the authors for completing such an interesting and valuable work.

However, the quality of the paper can be further improved, as commented as follows. Therefore, I would like to suggest a ‘Minor revision’. I would like to wish the authors every future success with their research and I hope my comments are useful to them.

1. Abstract. The significance and novelty of this research should be clarified.

2. Abstract. It is suggested to include some quantitative results.

3. A nomenclature is suggested.

4. Introduction. The objective of this study should be directly stated, with the description of research gap and motivation.

5. Introduction. The literature survey in the present version is not thorough, while some important studies on the modeling of district heating networks are missed.

6. Introduction. Following the above comment, without clear statement of research gap, the motivation and significance of this study cannot be evaluated.

7. Sections 2&3. These sections are structured well with useful information, but it seems a bit lengthy, in my own opinion.

8. Section 4. The demonstration section is well written and the results are solid. I have no comments on this part itself. Nevertheless, I believe it will be better if a discussion on the limitation of the model or this study can be included, to promote a better understanding for readers on this work. Although a paragraph mentioning the weakness of the model has been included in section 5, it cannot directly link to the results of this study. 

Reviewer 3 Report

Comments and Suggestions for Authors

Dear authors, thank you for your work. It is valuable input in the common Modelica society who dealing with district heating networks. Your work is well structured and clearly presented. Before publishing this work it is recommended to extend introduction section with short overview of the similar simulation programs which are appropriate for solving similar problems referencing actual studies, e.g. DOI 10.1016/j.scs.2022.103920, 10.1016/j.enbuild.2020.110538 etc. It will increase value of your study and gives balanced view on the available software programs. 

Round 2

Reviewer 2 Report

Comments and Suggestions for Authors

The authors well addressed the comments on the original version. At present form, I think the submission can be accepted for publication.